# The Level of Selected Blood Parameters in Young Soccer Players in Relation to the Concentration of 25(OH)D at the Beginning and End of Autumn

**DOI:** 10.3390/biology12010129

**Published:** 2023-01-13

**Authors:** Joanna Jastrzębska, Maria Skalska, Łukasz Radzimiński, Guillermo F. López Sánchez, Katja Weiss, Beat Knechtle

**Affiliations:** 1Department of Pediatrics, Diabetology and Endocrinology, Gdansk Medical University, 80-210 Gdansk, Poland; 2Department of Health and Natural Sciences, Gdansk University of Physical Education and Sport, 80-336 Gdansk, Poland; 3Division of Preventive Medicine and Public Health, Department of Public Health Sciences, School of Medicine, University of Murcia, 30120 Murcia, Spain; 4Institute of Primary Care, University Hospital Zurich, 8091 Zurich, Switzerland; 5Medbase St. Gallen Am Vadianplatz, 9001 St. Gallen, Switzerland

**Keywords:** football, seasonal variation, vitamin D, blood count, lipid profile, periodization

## Abstract

**Simple Summary:**

This study aimed to demonstrate the changes of selected blood parameters in relation to 25(OH)D concentration during autumn period in 35 young soccer players. The results presented in the current research showed that the level of 25(OH)D concentration significantly decreased after the autumn season in young soccer players. The declining level of 25(OH)D concentration should be compensated (e.g., with vitamin D supplementation) during autumn. Applied training loads could also influence the blood parameters variability in young soccer players.

**Abstract:**

This study aimed to demonstrate the changes of selected blood parameters in relation to 25(OH)D concentration during the autumn period in young soccer players. A total of 35 participants’ results (age: 17.5 ± 0.6 years, body mass 71.3 ± 6.9 kg) were tested twice: in mid-September and in mid-December and divided into subgroups with regard to two criteria. First, according to the initial level of the 25(OH)D concentration (optimal group—ODG, suboptimal group—SDG), second, according to drops in 25(OH)D concentration (high drop group—HDG, low drop group—LDG). A significant decrease (*p* < 0.001) in the 25(OH)D concentration was reported in the total group (TGr) and in all subgroups. Blood parameters such as white blood cells, red blood cells, haemoglobin and haematocrit increased significantly (*p* < 0.05) in TGr during the analysed period of time. The analysis of changes in the lipid profile did not expose significant differences except triglycerides. The asparagine amino transferase and creatine kinase activity decreased significantly after autumn in all analysed groups. The declining level of 25(OH)D concentration should be compensated (e.g., with vitamin D supplementation) during autumn. Applied training loads could also influence the blood parameters variability in young soccer players. Regular measurements of 25(OH)D concentration are helpful in identifying potential drops and allows for the preparation of individual supplementation plans for the players.

## 1. Introduction

The influence of vitamin D on human metabolic functions became a very popular scientific topic explored by researchers around the world. The significant effect of vitamin D on immune, anti-inflammatory, anticoagulant processes and cell death prevention was reported in past studies [1].

The common deficiency of vitamin D in Central and Northern Europe countries was established by multiple researchers [2,3]. Some authors [4,5] observed that an insufficient vitamin D level might lower physiological functions and physical fitness. Available reports confirmed that fitness components such as strength [6,7,8] and speed [9,10] are particularly affected, while the impact on aerobic capacity is still not fully recognized [11,12]. Fewer studies investigated the potential relations between the level of vitamin D associated with sunlight exposure, supplementation or nutrition and the risk of cardiovascular diseases [13,14], bone resorption [15], erythropoiesis [16], cancer risk [17] or multiple sclerosis [18]. De la Guia-Galipienso et al. [19] suggested that optimal levels of 25(OH)D concentration in blood plasma decrease the risk of cardiovascular diseases. In their cross-sectional research, Jorde et al. [20] found that a lower 25(OH)D concentration was negatively related to plasma lipid profile. Moreover, the meta-analysis provided by Kelishadi et al. [21] exhibited that an appropriate vitamin D level results in a favourable lipid profile in youth people. Furthermore, Monlezun et al. [22] reported that vitamin D deficiency might be related to an increased risk of anaemia and lower haemoglobin levels in healthy people. In addition, Patel et al. [23] observed that a lower level of vitamin D is related to chronic kidney disease [23]. Atkinson et al. [24] presented similar results for children and youths, while different findings were demonstrated by Ernst et al. [25], who supplemented hypertensive patients without anaemia with vitamin D (2800 IU/day) for eight weeks. This supplementation did not significantly affect the level of haemoglobin and anaemia risk.

Wilson-Barnes et al. [26] suggested that testing calcium (Ca), phosphorus (P) and parathormone (PTH) is particularly important when 25(OH)D concentration is below optimal values (30 ng/mL). Such a situation usually occurs during autumn and winter in countries with low ultraviolet (UV) radiation or in people who do not consume large amounts of vitamin D [27]. Brzeziański et al. [28] demonstrated that 8-week vitamin D supplementation (20 000 IU twice a week) is effective in 25(OH)D concentration compensation and allows for maintaining the appropriate level of PTH, leading to intensive bone reconstruction in young soccer players. The authors concluded that due to the complexity of the feedback mechanisms in maintaining proper calcium-phosphate homeostasis, determining the optimal dose of vitamin D supplementation and athletes’ training load is an important scientific topic. Furthermore, Bouassida et al. [29] reported that physical effort is a significant PTH concentration modifier depending on numerous factors, such as intensity, duration, type of exercise, age, sex, physical fitness, and initial bone mineral content. Therefore, analysing the training load can support a reliable assessment of the influence of vitamin D on the potential changes in blood parameters and physical capacity [30]. Interesting conclusions were presented by Jorde et al. [31], who showed that a 4-week vitamin D supplementation (20 000 IU a week) does not affect the level of bone resorption and bone turnover indicators in people without a 25(OH)D deficit. Thus, it can be assumed that only subjects with optimal or high levels of 25(OH)D concentration should participate in this type of scientific project.

The main purpose of this research was to demonstrate the changes of selected blood parameters in young soccer players in relation to the 25(OH)D concentration changes during the autumn season. It was hypothesized that a low level of 25(OH)D concentration and its’ dynamic decrease might negatively affect haematological blood parameters, lipid metabolism, and skeletal muscle damage. Moreover, we assumed that applied training loads could be considered an important factor that modifies changes in measured blood parameters.

## 2. Methods

### 2.1. Study Design

The research took place during a competitive autumn season lasting from mid-September 2019 (end of summer) to mid-December 2019 (end of autumn). At this time, the involved young soccer players should be optimally prepared for the league matches and maintain this level for three months. Before the research, it was assumed that the degree of insolation in the athletes’ place of residence from the beginning to the end of the research project would be reduced and would not affect the athletes’ vitamin D synthesis.

All the respondents performed the same soccer training program, including exercises of special techniques, tactics, endurance, speed, and muscle strength (Table 1). During the period of examination, training content differed in subsequent microcycles, depending on the tactics of the game and the phases of deliberate increase or decrease in training loads, taking into account, among other things, the effects of post-workout fatigue. Before and immediately after the testing phase, the players’ blood was drawn to determine the 25(OH)D concentration level and selected biochemical indicators. In addition, the number of UV index in each day of the project was documented. We also calculated athletes’ diet during the three months using a protocol for each player. Statistical analysis of the results was conducted on the whole test group (TGr) and divided into sub-groups. According to the median value of 25(OH)D concentration (32.3 ng/mL) obtained in the first test, individuals were divided into an optimal 25(OH)D concentration group (median (ME) < ODG) and a suboptimal 25(OH)D concentration group (Me > SDG). Another group division was made according to the changes (drops) in concentration of 25(OH)D in blood serum completion (individual changes before and after the project). Players were split into a high drop group (Me drop < HDG) and a low drop group (Me drop > LDG).

A training volume (total time in minutes) of a typical microcycle implemented during the research project is shown in Table 1. In the morning, players conducted team exercises of football techniques and tactics and upgraded their exercise capacity. In the afternoon, three days a week, the players trained individually in groups of goalkeepers, defenders, midfielders and forwards (formations). Thus, a typical weekly training load contained eight training sessions and one league game (12 h a week). The training experience of the study participants was between 10 and 12 years.

### 2.2. Participants

A total of 40 young soccer players from the same club took part in the study. The training experience of the study participants was between 10 and 12 years. All the participants competed in the Polish elite youth league. A total of 28 players lived in the club/school dormitory and received similar nutrition. The other 12 players lived in their homes. None of the participants consumed vitamin supplements from one month before the start and during the research. According to the Helsinki Declaration, the athletes, or their parents (in the case of underage players) signed the written consent and received detailed information about the project. Moreover, the study was approved by the Local Bioethics Medical Committee in Gdansk (Poland) (agreement number: 26/19).

### 2.3. Inclusion Criteria

We assumed that all the players were members of the same club and performed identical training loads. Only participants who completed at least 85% of training sessions were included in the study. Moreover, players who used another type of diet than the one preferred by the club; supplemented vitamin D or minerals, especially Ca and P; used the solarium; or trained outside without the allowance of the club trainers were excluded from the research. A total of 5 players did not meet the inclusion criteria (2 players were injured, another 2 missed more than 15% of training sessions, and 1 player was excluded due to illness). Finally, 35 players were included in the analysis of the results. The anthropometric characteristics of the tested players are introduced in Table 2, while the graphical presentation of the participants flowchart is showed in the Figure 1.

### 2.4. Procedures

#### 2.4.1. Degree of UV Radiation

The degree of ultraviolet was collected from the online weather service [32] that stores the historical data of the radiation. The study was performed in the city of Gdynia (Poland) between September and December 2019. The UV index scope in the analysed period of time is presented in Figure 2.

#### 2.4.2. Calculation of 25(OH)D and Biochemical Analyses

Biochemical analysis of the blood was conducted in an accredited analytical laboratory Diagnostics-medical laboratories, Gdynia, Poland). The Sysmex XE 2100 D and XT 4000 analysers (Sysmex Europe GmBH, Norderstedt, Germany) were used to calculate selected whole blood parameters. WBC (White Blood Cells), RBC (Red Blood Cells), HGB (Haemoglobin), HCT (Haematocrit), PLT (Thrombocytes) were calculated by fluorescent flow cytometry technology using reagents: Cellpack, Cellsheat, Stromatolyser FB, Strolamtolyser-4DL, Stromatolyser-4DS, Sulfolyser. The intra-assay CV of the method and with respect to the range were, respectively for: WBC 3,0% (at least 4.0 × 103/μL) and 4.0–10.0 10^3^/µL; RBC 1.5% (at least 4.00 × 106/μL) and 4.2–5.6 10^4^/µL; HGB 1,0% and 12.1–16.6 g/dL; HCT 1.5% and 35.0–49.0%; PLT 4.0% (at least 100 × 103/μL) and 18–>430 10^3^/μL.

TC (total cholesterol) was calculated using calorimetry with the cholesterol esterase reagent (Roche Cobas 6000). The intra-assay CV of the method was 1.6% for the range 115.0–190.0 mg/dL. HDL-C (high-density cholesterol) was calculated using calorimetry with the HDLC4 reagent (Roche Cobas 6000). The intra-assay CV of the method was 2.2% for the range ≥40 mg/dL. LDL-C (low-density cholesterol) was calculated by spectrophotometry with the LDLC3 reagent (Roche Cobas 6000). The intra-assay CV of the method was 2.5% for the range 0–130 mg/dL. TG (triglycerides) were calculated using calorimetry with the Roche 11-2017 reagent (Roche Cobas 6000). The intra-assay CV of the method was 2.0% for the range 0–150 mg/dL. 

ALT (alanine amino transferase) was calculated by spectrophotometry with the Roche 09-2018 reagent (Roche Cobas 6000). The intra-assay CV of the method was 3.3% for the range 0–41 U/L. AST (asparagine amino transferase) was calculated by spectrophotometry with the Roche 09-2018 reagent (Roche Cobas 6000). The intra-assay CV of the method was 2.3% for the range 0–40 U/L.

CK (creatine kinase) was calculated by spectrophotometry with the Roche 05-2017 reagent (Roche Cobas 6000). The intra-assay CV of the method was 3.2% for the range 39–308 U/L. 

The Chemiluminescence (CMIA) (Liaison XL, DiaSorin, Saluggia, Italy) using 25OH Vitamin D Total Assay reagent was applied to calculate the serum concentration of 25(OH)D. The intra-assay CV of the method was 2.4–6.4%, with respect to the range; 0–20 ng/mL deficit, >20–30 ng/mL suboptimal concentration, >30–50 ng/mL optimal concentration, >50–100 ng/mL high, >100 ng/mL potentially toxic, >200 ng/mL toxic [33].

#### 2.4.3. Calculation of Average Vitamin D Intake

The Dieta V 6.0 software (Poland, 2018) was used to assess the daily content of vitamin D in the food products eaten by every single player. This calculator was applied for a week, twice during the experiment. The first measurement took place during the first week (in September) and the second during the last week (in December) of the study. 

### 2.5. Statistical Analyses

The Shapiro–Wilk test was used for verifying the compliance of the normal distribution. PRE-POST differences were calculated with the t test for dependent trials (normal distribution data) or the Wilcoxon Paired Test (variables incompatible with normal distribution). A two-factorial variance analysis (ANOVA) was performed for repeated measurements to identify potential differences between the subgroups. The partial eta square (pη2) was calculated to demonstrate the effect size (ES). ES was defined as small (≥0.01), medium (≥0.06) and large (≥0.14) [34]. The significance level was set at *p* < 0.05, *p* < 0.01 or *p* < 0.001. All the analyses were performed using the software of Statistica 13.0 (TIBCO Software Inc., 2017, Palo Alto, CA, USA).

## 3. Results

The training volume in a typical training microcycle was 720 min (recovery periods covered 10% of the total training time). Aerobic intensity exercises accounted for 53.5% and mixed aerobic-anaerobic exercises for 34.7%. Additionally, soccer-specific drills constituted 61.1% of total training time (Table 1). 

The scope of ultraviolet (UV) radiation during the analysed period is presented in Figure 2. At the end of the summer, 77% of the participants had an optimal level of 25(OH)D concentration, whereas 23% had a suboptimal level. After the experiment, only 14% of the players had an optimal level, 57% had a suboptimal level and 29% had a vitamin D deficit. A significant decrease in 25(OH)D concentration was noted in the TG and all sub-groups (Figure 3).

The analysis of the haematological blood parameters indicated that only PLT did not change significantly during the autumn season. Moreover, the changes in lipid metabolism parameters reached the significance level only in TC (*p* < 0.001). In addition, a significant decrease in ALT and CK was noted at the end of the experiment (Table 3). 

The haematological blood parameters in the second measurement in SDG significantly increased in RBC, HGB and HCT compared to the first test. In contrast, in ODG, only WBC significantly increased. No significant changes in the lipid profile were noted in the analysed period except SDG, where TG increased significantly (*p* < 0.01). Moreover, both groups reported large ES and significant AST and CK reductions. Finally, the significant group x time interaction (ES = 0.13) was reported for RBC (Table 4). 

Table 5 shows the results from the beginning and the end of the autumn seasons after dividing the players into HDG and LDG. Despite PLT, lipid profile, and ALT, the significant (ES > 0.14) time interaction was registered for all parameters. The significant pre-post changes were noted in five parameters in HDG and two parameters in LDG.

## 4. Discussion

Vitamin D deficiency is common in the countries of Northern Europe. During autumn, winter and early spring, vitamin D demand is mainly covered by food sources and supplementation because its natural synthesis is very low. This problem is especially visible in young athletes exposed to large training loads. This research was projected to investigate the changes in 25(OH)D concentration in young soccer players from Northern Europe (Poland) in the autumn season when the sunlight radiation decreases, and vitamin D synthesis is very low. Moreover, it was demonstrated how the initial level and larger or smaller drop in 25(OH)D concentration influence the changes in selected blood parameters of tested athletes. Finally, the effect of applied training loads on these parameters was also evaluated.

The main finding of this research was that 25(OH)D concentration significantly decreased in TGr and all four subgroups during the analysed period of time. Based on the statistical calculations, this reduction could significantly affect activity or changes in several blood parameters of young soccer players. Considering the interaction between the groups and time, the large ES was reported when comparing pre and post measurements. In addition, it was concluded that despite the changes in 25(OH)D concentration, the applied training loads could also influence the blood parameters variability (Figure 3).

Numerous factors, such as UV radiation, daytime, the amount of skin melanin, diet, and supplementation have been recognized as variables affecting the natural synthesis of vitamin D [33]. Wearing clothes covering most of the body or anti-radiation creams result in limited level 25(OH)D concentration [35]. Current research shows that the average level of UV season radiation was registered in September during the autumn, while low and very low levels were noted from October to December. Sieniawska et al. [36] demonstrated that the average UV radiation in Northern Poland (54° N) during summertime (26 June–7 July) was 7.3 (from 1.6 to 10.1 a day). At this time, the 25(OH)D concentration in children from that region increased from 20.3 ng/mL to 26.2 ng/mL. Thus, it can be assumed that our study participants could obtain vitamin D in endogenous transformation only to a small extent because the UV radiation value did not exceed 3.5 in September and fell even lower in the following month (Figure 2).

The analysis of participants’ diet indicated that vitamin D content in supplied products was insufficient to secure its daily dose. The daily vitamin D intake of 11 players living at home was 165.35 ± 18.52 IU/day in the first week and 179.46 ± 14.42 IU/day in the last week of the study. At the same time, 24 participants from the dormitory received 145.21 ± 18.52 IU/day and 155.46 ± 15.47 IU/day, respectively. These values were several times lower than 2000 IU/day, which is recommended dose for Polish youth during autumn and winter [37,38]. Backx et al. [39] presented similar advice, who proposed a dose of 2200 IU/day as effective in obtaining the optimal level of 25(OH)D concentration. Moreover, these authors found that tested athletes provided an average of 168 ± 104 IU/day of vitamin D, which is in line with our research values. Brustad et al. [40] suggested that such low vitamin D intake could result from a diet low in fish, fish and vegetable oils, and the lack of nutritional prevention containing vitamin D supplementation.

Our previous publication on the same group of young soccer players analysed the changes in physical fitness and bone resorption markers [41]. The level of aerobic capacity significantly improved at the end of the autumn season, while aerobic capacity levels remained constant. Significant changes in bone resorption markers were noted at this time as well.

In the present study, a significant (*p* < 0.05) decrease in the level of 25(OH)D concentration was registered at the end of the autumn season for all the analysed groups (Figure 2). These results align with Bezuglov et al. [42] who analysed 131 young soccer players from similar latitude (55° N) and found that in the December the 25(OH)D concentration was 19.7 ± 5.4 ng/mL. Furthermore, Serdar et al. [15] made a yearly observation of 13026 untrained Turkish women and men. The largest and significant drops of 25(OH)D concentration was reported in both genders between September (29.52 ± 16.0 ng/mL) and December (24.2 ± 17.2 ng/mL), what is in line with our results.

Our research results showed that despite the PLT, all the morphological blood parameters significantly increased in TG after the autumn season (WBC 14.83%, RBC 2.86%, HGB 3.29%, HCT 2.57%). Muller et al. [43] suggested that the initial level of 25(OH)D concentration could influence changes caused by the different times of the year or supplementation. According to Yagüe et al. [30], supplementation had more significant, positive effects on people with vitamin D deficiency at the initial stage. A greater number of changes in morphological blood parameters were registered in our experiment according to RBC (4.32%), HGB (4.64%) and HCT (3.79%) in the group with the lower initial level of 25(OH)D concentration (SDG). In contrast, in ODG significant increase in post-experiment measurements were noted only in WBC (13.02%). Due to the contribution to aerobic metabolism, the higher values of the HGB index seem to be beneficial for young soccer players. 

Furthermore, after dividing the players into HDG and LDG, the significant differences were noted only in WBC and HGB in the group with higher drops of 25(OH)D. Thus, it can be concluded that all haematological indicators results except WBC improved in the post-experiment measurement for all the groups. The significant increase in WBC could suggest the immunological defence against the pro-inflammatory cytokines [44]. This reaction seems to be possible due to the large number of high-intensity efforts applied to the players during analysed period of the season. Hashemi et al. [45] observed that high levels of vitamin D in the human body increase the activity of anti-inflammatory interleukins IL-27, TGF-β1 and IL-10, and decrease the activity of pro-inflammatory IL-17A, IL-6. Moreover, Verdoia and Da Luca [46] registered similar relations with reference to acute phase proteins (CRP). The decrease in 25(OH)D concentration in tested players could limit the immune reaction and result in a significant increase in WBC. The results of this study did not confirm this statement in relation to RBC and HGB because the low level of 25(OH)D concentration at the beginning of the project and its large drop did not limit the significant improvements and better cellular oxygenation. These changes were rather caused by the large training volume of analysed players then by unfavourable changes in 25(OH)D concentration [47]. Similar conclusions were presented by Skalska et al. [48], who tested soccer players from the same age group and suggested that training loads may have a stronger effect on youth athletes’ bodies than seasonal 25(OH)D changes in blood plasma.

The mechanism regulating lipid profile changes in people supplemented with vitamin D is ambiguous. Although Wang et al. [49] reported in their meta-analysis that a higher level of vitamin D may result in increased LDL cholesterol fraction and has no significant effect on TC, HDL and TG, the review paper of Dibaba [13] demonstrated that vitamin D supplementation might increase the level of TC, LDL, TG and has no influence on HDL fraction. Furthermore, Kelishadi et al. [21] indicated that a higher level of 25(OH)D is related to a favourable lipid profile in the paediatric age group. In contrast, Jastrzębska et al. [41] demonstrated no significant effect of 8-week vitamin D supplementation on the haematological parameters and lipid profile during the pre-season period. The lack of significant pre-post changes in lipid profile (except TG) was shown in all groups. The significant increase in TG could be potentially associated with larger carbohydrates intake during the period of higher training loads [50].

Analysing the muscle damage blood parameters in tested soccer players, the most significant changes were found in CK. In TG, the value of this variable decreased to 216.9 U/L. Taking into consideration the initial concentration of 25(OH)D, a more significant reduction was observed in ODG (−237.83 U/L) and HDG (−256.83 U/L). Żebrowska et al. [38] observed that a higher level of 25(OH)D after vitamin D supplementation (2000 IU/day) increases the preventive effect of eccentric exercises and protects against muscle damage. The lower values of CK at the end of the autumn were probably caused by training adaptation to the changes in 25(OH)D concentration. This thesis is supported by the presented results, where a significant reduction in CK was noted for all groups. Similar conclusions were presented by Meyer and Meister [51] in their review and meta-analysis study. The authors suggested that high-intensity interval training in soccer and match competition significantly affects HCT index characterizing blood plasma volume and CK activity.

The activity of such liver and muscles damage markers as ALT and AST decreased during the analysed period of time. However, only AST changes reached the significance level. Javed et al. [52] reported that after 3-month supplementation of vitamin D (3200 IU/day), a higher concentration of 25(OH)D could be related to a significant reduction of AST in tested women. Due to the decrease in 25(OH)D recorded in the post-experiment measurement, the lower activity of liver enzymes is probably related to young soccer players’ training adaptation than negative 25(OH)D changes.

Although the presented research was conducted on a group of young soccer players performing identical training loads and similar nutrition habits, some limitations of the study are worth mentioning. A larger number of participants would undoubtedly strengthen the power of statistical analysis. However, a larger sample is difficult to obtain due to the specifics of soccer teams that usually involve about 25 players. Nevertheless, future research should be conducted on larger groups and investigate possible changes in 25(OH)D concentration and blood parameters in relation to applied training loads. During the implementation of our project, we observed that such research is related with some limitation. Lack of possibility of the continuous training monitoring could be considered as a problem. Coaches do not always agree with significant interference in the training process. Furthermore, longitudinal research on team sports athletes is a challenge as well due to injuries occurrence or changing clubs by the players. Another important task was to control the diet of the participants, especially of those who lived with their families. The most recommended solution is to analyse participants living in the dormitory where number and content of the meals could be controlled by a nutritionist. 

The results of our research contain high application values for sport practitioners. It could be used during the training process in both, team and individual sports. Although the topic of changes in 25(OH)D concentration has been widely described, the interpretation of these data with reference to training cycles and appropriate seasons is still scarce. Current study shows the dynamics changes in 25(OH)D concentration in young soccer players as a result of lower natural vitamin D synthesis. Therefore, such information suggests that vitamin D supplementation should be applied to prevent its deficiency. Insufficient levels of vitamin D could lead to negative changes in blood biomarkers and decrease physical fitness. Another important practical aspect of our research was to present the changes in selected blood biomarkers in players with greater and lower drops of 25(OH)D concentration. Such an approach enables individual supplementation of vitamin D. Our experience suggests that supplementation should be applied from half of October to half of April. Doses between 2000 IU/day and 4000 IU/day seem to be appropriate. We encourage coaches to cooperate with scientists investigating the topics of seasonal changes in 25(OH)D concentration. New projects involving homogenous groups of athletes could provide additional observations. 

## 5. Conclusions

The results presented in the current research showed that the level of 25(OH)D concentration significantly decreased after the autumn season in young soccer players. These changes were caused mainly by the low level of UV radiation and insufficient vitamin D intake in the players’ diet. The significant changes in haematological blood parameters were found in TGr except for PLT. Moreover, more significant differences were observed in SDG and HDG. Except for WBC, these changes were positive regarding the aerobic metabolism, which can be associated with training adaptation than with seasonal changes in 25(OH)D concentration. Analysis of changes in lipid profile did not expose significant differences except TG. Increased value of triglycerides could result from increased intake of simple carbohydrates during the period of high-intensity training loads during the competitive season. The significant decrease in AST and CK activity in all analysed groups after the autumn season was caused by the beneficial effect of applied training loads rather than the reduction of 25(OH)D concentration in blood plasma.

## Figures and Tables

**Figure 1 biology-12-00129-f001:**
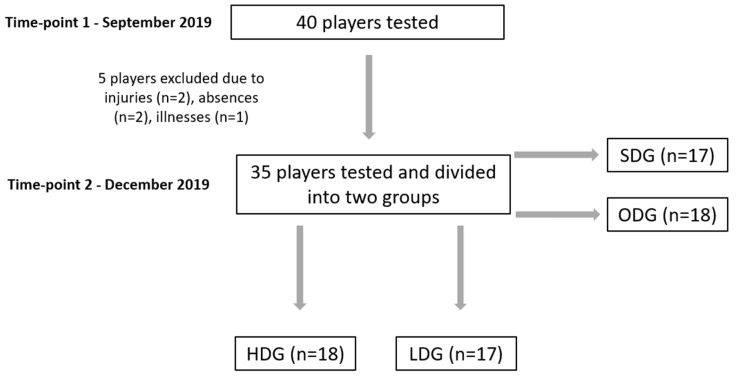
Presentation of the study design and flowchart of the participants. SDG- suboptimal 25(OH)D concentration group; ODG- optimal 25(OH)D concentration group; HDG- high drop group; LDG- low drop group.

**Figure 2 biology-12-00129-f002:**
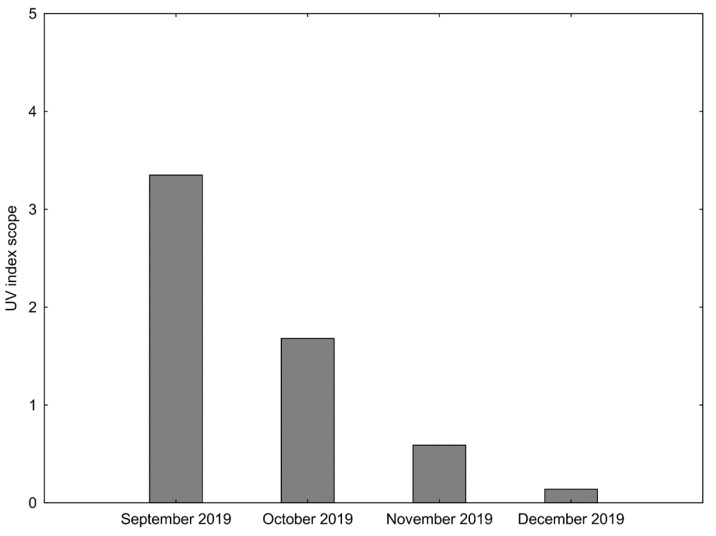
The scope of ultraviolet (UV) radiation during the analysed period of time [https://www.weatheronline.co.uk, access on 15 February 2020].

**Figure 3 biology-12-00129-f003:**
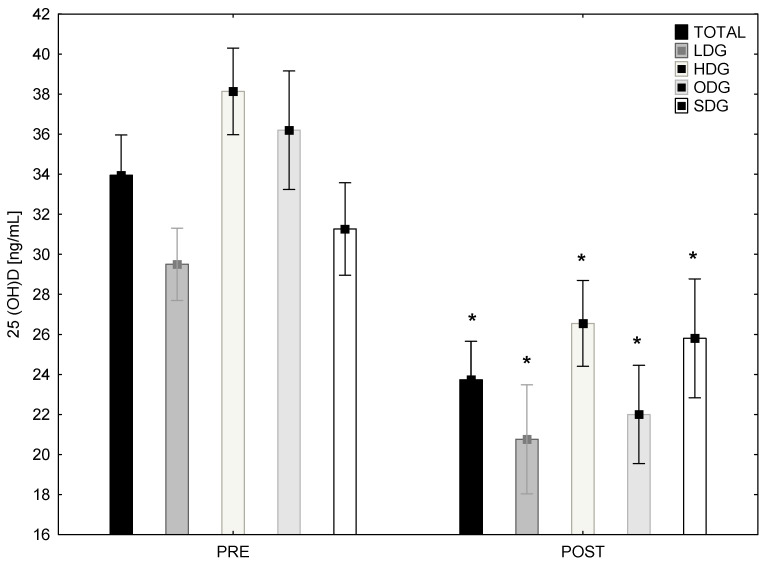
Level of 25(OH)D concentration in blood plasma in analysed young soccer players at the beginning and the end of the autumn in TOTAL, LDG, HDG, ODG and SDG. * significantly different from PRE at *p* < 0.001. TG-total group; SDG—suboptimal 25(OH)D concentration group; ODG—optimal 25(OH)D concentration group; HDG—high drop group; LDG—low drop group.

**Table 1 biology-12-00129-t001:** Training load (min) taking into account type (overall and special drills) and intensity (aerobic, anaerobic) of exercises of typical weekly training during competition period of young soccer players.

	**Aerobic Performance (min)**	**Aerobic-Anaerobic Performance (min)**	**Anaerobic Lactate Performance (min)**	**Anaerobic Non lactate Performance (min)**	**Total (min)**
Overall drills	225	30	5	20	280
Special drills	160	220	30	30	440
Total (min)	385	250	35	50	720

**Table 2 biology-12-00129-t002:** Anthropometric characteristics of the study participants.

Group/Variable	Age [years]	Body Mass [kg]	Height [cm]	BMI [kg/m^2^]
TG (n = 35)	17.5 ± 0.6	71.3 ± 6.9	178.9 ± 4.84	22.2 ± 1.8
ODG (n = 18)	17.4 ± 0.6	70.4 ± 5.9	179.3 ± 5.11	21.9 ± 2.0
SDG (n = 17)	17.6 ± 0.5	72.0 ± 7.4	178.3 ± 6.74	22.6 ± 1.6
HDG (n = 18)	17.4 ± 0.5	71.8 ± 6.8	179.9 ± 7.65	22.2 ± 1.9
LDG (n = 17)	17.6 ± 0.7	71.0 ± 6.8	178.0 ± 5.03	22.4 ± 1.6

TG—total group; SDG—suboptimal 25(OH)D concentration group; ODG—optimal 25(OH)D concentration group; HDG—high drop group; LDG—low drop group.

**Table 3 biology-12-00129-t003:** Changes of selected blood parameters in total group (TGr n = 35).

**Variable**	**PRE**	**POST**	* **p** *
WBC (10^3^/µL)	5.93 ± 1.05	6.61 ± 1.33 *	0.0047
RBC (10^4^/ µL)	5.14 ± 0.29	5.29 ± 0.35 *	0.0008
HGB (g/dL)	15.0 ± 0.66	15.51 ± 0.73 *	0.0001
HCT (%)	44.78 ± 1.90	45.96 ± 2.14 *	0.0005
PLT (10^3^/µL)	233.6 ± 39.06	236.6 ± 37.18	0.2918
TC (mg/dL)	140.9 ± 20.70	143.1 ± 20.43	0.3570
HDL-C (mg/dL)	56.3 ± 9.24	55.7 ± 8.62	0.6284
LDL-C (mg/dL)	71.8 ± 18.37	68.2 ± 15.64	0.1502
TG (mg/dL)	63.7 ± 20.79	94.1 ± 44.21 *	0.0000
ALT (U/L)	19.5 ± 5.40	19.4 ± 7.75	0.3469
AST (U/L)	29.2 ± 8.47	23.0 ± 5.59 *	0.0000
CK (U/L)	407.3 ± 236.28	190.4 ± 110.80 *	0.0000

* PRE-POST differences at *p* < 0.05.

**Table 4 biology-12-00129-t004:** Changes in selected blood biochemical and physical performance indicators in young footballers in the group with optimal (ODG, n = 18) and suboptimal (SDG, n = 17) 25(OH)D blood serum concentration before (PRE) and after (POST) testing program.

	**ODG (n = 18)**		**SDG (n = 17)**			
**Variable**	**PRE**	**POST**	**PRE**	**POST**	**Interaction**	**ES**
WBC (10^3^/µL)	5.61 ± 0.69	6.45 ± 0.93 †	6.26 ± 1.26	6.77 ± 1.66	Time	0.21
RBC (10^4^/ µL)	5.18 ± 0.29	5.25 ± 0.35	5.10 ± 0.29	5.33 ± 0.35 ^#^	TimeGroup x time	0.320.13
HGB (g/dL)	15.19 ± 0.43	15.48 ± 0.75	14.81 ± 0.81	15.53 ± 0.74 ^#^	Time	0.41
HCT (%)	45.32 ± 1.57	45.97 ± 1.98	44.21 ± 2.09	45.95 ± 2.36 ^#^	Time	0.33
PLT (10^3^/µL)	221.89 ± 33.99	230.22 ± 34.38	245.94 ± 41.22	242.76 ± 38.93	-	-
TC (mg/dL)	144.44 ± 19.23	146.56 ± 19.54	137.06 ± 22.08	139.47 ± 21.31	-	-
HDL-C (mg/dL)	56.22 ± 9.15	54.44 ± 9.19	56.35 ± 9.61	57.00 ± 8.04	-	-
LDL-C (mg/dL)	74.78 ± 15.98	71.72 ± 12.74	68.65 ± 20.62	64.41 ± 17.84	-	-
TG (mg/dL)	67.56 ± 26.50	102.56 ± 49.94	59.53 ± 11.71	85.24 ± 36.62 ^†^	Time	0.40
ALT (U/L)	20.83 ± 5.52	21.17 ± 9.17	18.06 ± 5.04	17.53 ± 5.57	-	-
AST (U/L)	30.78 ± 8.00	24.17 ± 5.42 ^†^	27.53 ± 8.88	21.76 ± 5.67 ^#^	Time	0.52
CK (U/L)	445.33 ± 253.51	207.50 ± 135.75 ^#^	367.12 ± 216.82	172.24 ± 76.33 ^#^	Time	0.52

PRE-POST differences at: † *p* < 0.01, # *p* < 0.001; ES—effect size; ODG—optimal 25(OH)D concentration group; SDG—suboptimal 25(OH)D concentration group; WBC—white blood cells; RBC—red blood cells; HGB—haemoglobin; HCT—haematocrit; PLT—thrombocytes; TC—total cholesterol; HDL-C—high-density cholesterol; LDL-C—low-density cholesterol; TG—triglycerides; ALT—alanine aminotransferase; AST—aspartate aminotransferase; CK—creatine kinase.

**Table 5 biology-12-00129-t005:** Changes in selected blood biochemical and physical performance indicators in young footballers in the group with high HDG (*n* = 18) and low (LDG) (*n* = 17) drops of 25(OH)D in blood serum concentration before (PRE) and after (POST) testing program.

	**HDG (n = 18)**		**LDG (n = 17)**			
**Variable**	**PRE**	**POST**	**PRE**	**POST**	**Interaction**	**ES**
WBC (10^3^/µL)	5.80 ± 0.98	6.77 ± 1.29†	6.08 ± 1.13	6.42 ± 1.40	Time	0.21
RBC (10^4^/ µL)	5.15 ± 0.29	5.3 ± 0.34	5.13 ± 0.30	5.3 ± 0.38	Time	0.28
HGB (g/dL)	15.21 ± 0.48	15.7 ± 0.66 *	14.76 ± 0.77	15.3 ± 0.80	Time	0.39
HCT (%)	45.02 ± 1.59	46.0 ± 1.82	44.50 ± 2.23	45.9 ± 2.53	Time	0.31
PLT (10^3^/µL)	230.63 ± 37.71	236.2 ± 39.81	237.06 ± 41.57	236.5 ± 33.84	-	-
TC (mg/dL)	137.21 ± 19.59	140.2 ± 22.20	145.19 ± 21.76	146.6 ± 18.22	-	-
HDL-C (mg/dL)	58.47 ± 9.49	57.3 ± 9.53	53.69 ± 8.49	53.8 ± 7.21	-	-
LDL-C (mg/dL)	67.00 ± 16.82	64.0 ± 14.01	77.50 ± 19.02	73.2 ± 16.41	-	-
TG (mg/dL)	58.79 ± 14.97	95.1 ± 47.64 ^†^	69.44 ± 25.40	93.0 ± 41.30	Time	0.40
ALT (U/L)	19.53 ± 4.71	21.4 ± 8.98	19.44 ± 6.29	17.0 ± 5.30	-	-
AST (U/L)	30.53 ± 7.95	23.7 ± 5.57 ^†^	27.63 ± 9.05	22.2 ± 5.69 *	Time	0.52
CK (U/L)	435.53 ± 265.12	178.7 ± 100.25 ^#^	373.88 ± 20.00	204.3 ± 124.04 ^†^	Time	0.52

PRE-POST differences at: * *p* < 0.05, † *p* < 0.01, # *p* < 0.001; ES—effect size; HDG—high drop group; LDG—low drop group; WBC-white blood cells; RBC—red blood cells; HGB—haemoglobin; HCT—haematocrit; PLT—thrombocytes; TC—total cholesterol; HDL-C—high-density cholesterol; LDL-C—low-density cholesterol; TG—triglycerides; ALT—alanine aminotransferase; AST—aspartate aminotransferase; CK—creatine kinase.

## Data Availability

Data supporting the findings of this study are available from the corresponding author upon reasonable request.

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
