# Peer review of "The Level of Selected Blood Parameters in Young Soccer Players in Relation to the Concentration of 25(OH)D at the Beginning and End of Autumn"

_biology, 2023, doi:10.3390/biology12010129_

Round 1

Reviewer 1 Report

PDF with comments is included

Author Response

We would like to thank the reviewer for her/his valuable advices. Below we addressed all the comments point by point. We strongly believe that all the modifications will improve the quality of our manuscript.

Comments to the Author

General comment

The following is a review of the article entitled "The level of selected blood parameters in young soccer players 2 in relation to the concentration of 25(OH)D at the beginning and end of autumn" which aims to: to demonstrate the changes of selected blood parameters in relation to 25(OH)D concentration during autumn period in young soccer players. Thank you very much for thinking of me as a reviewer for this study. After carefully reading the manuscript, I set forth comments and suggestions for the authors:

Abstract:

Suggestion - Correct, but the conclusions could be expanded. Add more practical application and some of the limitations found.

Answer: A short statement containing practical applications was implemented in the abstract according to Reviewer suggestion. Furthermore, in other part of the text we introduced wider description of limitation and practical application

Keywords:

Suggestion - The keywords are correct.

Answer – thank you

Introduction:

Suggestion - Generally, correct. However, some possible corrections are added. Replace "Jorge" by "Jorde" (line 82). Add practical applications and some of the limitations found.

Answer: The indicated reference has been fixed according to Reviewers’ suggestion.

Moreover, we introduced the text including practical applications and limitations of the study. However, we suggest to include this text in the final part of the Discussion section. We would be grateful if Reviewer could accept our proposition.

Limitations:

During the implementation of our project we observed that such research is related with some limitation. Lack of possibility of the continuous training monitoring could be considered as a problem. Coaches not always agree for significant interference in the training process. Furthermore, longitudinal research on team sports athletes is a challenge as well due to injuries occurrence or changing clubs by the players. Another important task was to control the diet of the participants, especially of those who lived with their families. The most recommended solution is to analyze participants living in the dormitory where number and content of the meals could be controlled by a nutritionist.

Practical applications: The results of our research contain high application values for sport practitioners. It could be used during the training process in both, team and individual sports. Although the topic of changes in 25(OH)D concentration has been widely described, the interpretation of these data with reference to training cycles and appropriate seasons is still scarce. Current study shows the dynamics changes in 25(OH)D concentration in young soccer players as a result of lower natural vitamin D synthesis. Therefore, such information suggest that vitamin D supplementation should be applied to prevent its’ deficiency. Insufficient level of vitamin D could lead to the negative changes in blood biomarkers and decrease the physical fitness. Another important practical aspect of our research was to present the changes in selected blood biomarkers in players with greater and lower drops of 25(OH)D concentration. Such approach enables individual supplementation of vitamin D. Our experience suggest that supplementation should be applied from half of October to half of April. Doses between 2000 IU/day and 4000 IU/day seem to be appropriate. We encourage coaches to cooperate with scientists investigating the topics of seasonal changes in 25(OH)D concentration. New projects involving homogenous groups of athletes could provide additional observations.

Materials and Methods:

Suggestion - Generally, correct. However, some possible corrections are added.

  1. Add information about this: " In addition, the number of sunny days and the athletes’ diet during the three months were documented" (line 109).
  2. How the diet of the 12 players living at home was controlled (line 141).
  3. Include a CONSORT diagram as a flowchart indicating the final sample (line 144).
  4. Add table with descriptive data (line 146).
  5. Could the experience of the participants be included? Shouldn't the biological age of the athletes have been controlled? It would be very important to know if the participants are in pre or post pubertal age.
  6. Could the information in “Figure 1” be completed? (line 154).
  7. More information about the section" Calculation of average vitamin D Intake" (line 190).

Answers :

  1. We changed the sentence to "In addition, the number of UV index in each day of the project was documented. We also calculated athletes’ diet during the three months using protocol for each player" (line 109). Moreover, in this part we described general study design, while further – in Procedure section – each measurement was introduced. We could demonstrate the number od sunny days as well, however, the UV index provide more information about the effect of UV rays on the natural synthesis of vitamin D, because it takes into account latitude the angle of sun’s rays) and intensity of the sun rays.
  2. Every player regardless if he was living at home or in the dormitory received protocol, where all the meals were described in details. Appropriate description was included as follows: “All the participants living in the dormitory or in their homes, received a protocol, where detailed number and content of the meals were described during consecutive days of the week (at the beginning and at the end of the experiment). The club nutritionist controlled the validity of individual protocols.” Furthermore, information about the used software is in the chapter 3.4.3.
  3. According to Reviewer suggestion, we’ve created a new Figure 1 where the flowchart of the participants is presented.
  4. The additional table involving the anthropometric characteristic of the players was inserted.
  5. We agree that both, training experience and biological age are important variables that should be taken into consideration in such analysis. The training experience of the tested players was introduced in the text (in the chapter 3.2). The biological age of the players was tested by the club doctor during regular medical exams. The routine practice in the club assume that during these exams doctor evaluate the puberty using the criteria described by Tanner [Tanner JM. Growth at adolescence. 2nd Ed. Oxfordshire: Blackwell Scientific Publications, 1962]. All the participants reached the 5th stage before beginning of the project.
  6. We are not sure if we understand the Reviewer suggestion correctly. Instead of presented figure we are able to present (if Reviewer find it necessary) the UV data in the daily form. Please see the presented below proposition.
  7. Additional information about players diet were introduced in point 2 and in Study design section.

Results:

Generally, correct. However, some possible corrections are added. Tables and figures should be interpreted without reading the text. They should contain all the information included in them (review all).

Answer – All the used abbreviations were explained below the tables so the readers could interpret the data without familiarization with all paper.

Discussion:

Generally, correct. However, some possible corrections are added.

  1. Is it possible to include data from references 42 and 15? (line 303).

Answer – According to Reviewer suggestions the data from references 42 and 15 were introduced in the text as follows:

In the present study, a significant (p<0.05) decrease in the level of 25(OH)D concentration was registered at the end of the autumn season for all the analysed groups (Figure 1). These results align with Bezuglov et al. [42] who analysed 131 young soccer players from similar latitude (55oN) and found that in the December the 25(OH)D concentration was 19.7±5.4 ng/mL. Furthermore, Serdar et al. [15] made a yearly observation of 13026 untrained Turkish women and men. The largest and significant drops of 25(OH)D concentration was reported in both genders between September (29.52±16.0 ng/mL) and December (24.2±17.2 ng/mL), what is in line with our results.

  1. It is possible to add references to support this statement: “The significant increase in WBC could suggest the immunological defence against the pro-inflammatory cytokines” (line 320).

Answer – The appropriate reference has been introduced (Jamurtas et al. 2018, doi: https://doi.org/10.3390/antiox7040050.

  1. Add practical applications.

Answer – The practical application was implemented in the text. Moreover, the limitation paragraph has been extensively developed according to the Reviewers advices.

Conclusions: Correct.

Answer – Thank you

References: Corrects

Answer – Thank you

Reviewer 2 Report

very interesting and actual topic. Only a few comments.

1. improve citation list- not current citation  (14), (19)- better :

Manson JE, Cook NR, Lee IM, Christen W, Bassuk SS, Mora S, Gibson H, Gordon D, Copeland T, D'Agostino D, Friedenberg G, Ridge C, Bubes V, Giovannucci EL, Willett WC, Buring JE; VITAL Research Group. Vitamin D Supplements and Prevention of Cancer and Cardiovascular Disease. N Engl J Med. 2019 Jan 3;380(1):33-44. doi: 10.1056/NEJMoa1809944. Epub 2018 Nov 10. PMID: 30415629; PMCID: PMC6425757.

 de la Guía-Galipienso F, Martínez-Ferran M, Vallecillo N, Lavie CJ, Sanchis-Gomar F, Pareja-Galeano H. Vitamin D and cardiovascular health. Clin Nutr. 2021 May;40(5):2946-2957. doi: 10.1016/j.clnu.2020.12.025. Epub 2020 Dec 29. PMID: 33397599; PMCID: PMC7770490.

2. page 2, line 62--the citation 23 is not from Monlezun, but other author, the same page 11, line 326 citation 45

3.page 3. line 114- first abbreviation Me, than whole word

4. results- the results of the dietary regime analysis are missing, there are in the discussion..

5. page 14, line 480- year of edition is missing

Author Response

We would like to thank the reviewer for her/his valuable suggestions. Below we addressed all the comments point by point. We strongly believe that all the modifications will improve the quality of our manuscript.

Comments to the Author

General comment

Very interesting and actual topic. Only a few comments.

Suggestion 1:

Improve citation list- not current citation  (14), (19)- better :

Manson JE, Cook NR, Lee IM, Christen W, Bassuk SS, Mora S, Gibson H, Gordon D, Copeland T, D'Agostino D, Friedenberg G, Ridge C, Bubes V, Giovannucci EL, Willett WC, Buring JE; VITAL Research Group. Vitamin D Supplements and Prevention of Cancer and Cardiovascular Disease. N Engl J Med. 2019 Jan 3;380(1):33-44. doi: 10.1056/NEJMoa1809944. Epub 2018 Nov 10. PMID: 30415629; PMCID: PMC6425757.

de la Guía-Galipienso F, Martínez-Ferran M, Vallecillo N, Lavie CJ, Sanchis-Gomar F, Pareja-Galeano H. Vitamin D and cardiovascular health. Clin Nutr. 2021 May;40(5):2946-2957. doi: 10.1016/j.clnu.2020.12.025. Epub 2020 Dec 29. PMID: 33397599; PMCID: PMC7770490.

Answer – We would like to thank the Reviewer for proposing newer research. Both publications were implemented in the manuscript.

Suggestion 2:

Page 2, line 62--the citation 23 is not from Monlezun, but other author, the same page 11, line 326 citation 45.

Answer – Indeed, we have corrected both statements. These inaccuracies arose during translation.  

Suggestion 3:

page 3. line 114- first abbreviation Me, than whole word

Answer – The text was modified according to the Reviewer suggestion as follows:

According to the median value of 25(OH)D concentration (32.3 ng/mL) obtained in the first test, individuals were divided into an optimal 25(OH)D concentration group (median Me<ODG) and a suboptimal 25(OH)D concentration group (Me>SDG). Another group division was made according to the changes (drops) in concentration of 25(OH)D in blood serum completion (individual changes before and after the project). Players were split into a high drop group (Me<HDG, drop) and a low drop group (Me>LDG, drop).

Suggestion 4:

Results- the results of the dietary regime analysis are missing, there are in the discussion.

Answer – Due to the fact that measurements of vitamin D intake included only 4 numbers and it fit well in the discussion, we decided to put these data in this chapter. Although we are aware that this is not a standard procedure, we suggest to leave the results in the current form.

Suggestion 5:

Page 14, line 480- year of edition is missing.

Answer – Corrected. Thank you for this observation.

Round 2

Reviewer 1 Report

Dear Authors, 

I appreciate the reviews. 

Best regards,